# Enhancing Prognosis in Advanced Ovarian Cancer: Primary Cytoreductive Surgery and Adjuvant Chemotherapy or Neoadjuvant Chemotherapy and Interval Cytoreduction—A Single-Center Retrospective Observational Study

**DOI:** 10.3390/cancers17081314

**Published:** 2025-04-14

**Authors:** Adelina Silvana Gheorghe, Irina Alexandra Chirea, Mădălin Marius Margan, Mihai-Teodor Georgescu, Isabela Anda Komporaly, Lidia Anca Kajanto, Elena Adriana Iovănescu, Bogdan Georgescu, Radu Matei, Daniela Luminița Zob, Mara Mardare, Octav Ginghină, Mara Mădălina Mihai, Dana Lucia Stănculeanu

**Affiliations:** 1Department of Medical Oncology I, “Prof. Dr. Alexandru Trestioreanu”, Institute of Oncology, 022328 Bucharest, Romania; adelina.silvana.gheorghe@gmail.com (A.S.G.); irina.ic.ii@gmail.com (I.A.C.); lidia.kajanto@gmail.com (L.A.K.); dumitrescu.elena.adriana@gmail.com (E.A.I.); radu.mateirm@yahoo.com (R.M.); dlstanculeanu@gmail.com (D.L.S.); 2Department of Functional Sciences, Discipline of Public Health, Victor Babes University of Medicine and Pharmacy, 300041 Timisoara, Romania; 3Department of Oncology, Faculty of Medicine, “Carol Davila” University of Medicine and Pharmacy, 020021 Bucharest, Romania; isabelakomporaly@gmail.com (I.A.K.); dr_bogdan.georgescu@icloud.com (B.G.); 4Department of Radiotherapy II, “Prof. Dr. Alexandru Trestioreanu”, Institute of Oncology, 022328 Bucharest, Romania; 5Memorial Hospital, 013812 Bucharest, Romania; 6Neolife Hospital, 077190 Bucharest, Romania; 7Department of Medical Oncology II, “Prof. Dr. Alexandru Trestioreanu”, Institute of Oncology, 022328 Bucharest, Romania; danielazob@yahoo.com; 8Department of Surery III, “Prof. Dr. Alexandru Trestioreanu”, Institute of Oncology, 022328 Bucharest, Romania; mara_mardare@yahoo.com (M.M.); octav.ginghina@umfcd.ro (O.G.); 9Department of Surgery, Faculty of Dentistry, “Carol Davila” University of Medicine and Pharmacy, 020021 Bucharest, Romania; 10Department of Oncologic Dermatology, “Carol Davila” University of Medicine and Pharmacy, 020021 Bucharest, Romania; mara.mihai@umfcd.ro; 11Elias Emergency Universitary Hospital, 011461 Bucharest, Romania

**Keywords:** ovarian cancer, primary cytoreductive surgery, neoadjuvant chemotherapy, interval debulking surgery, progression-free survival, overall survival, BRCA mutations, homologous recombination deficiency, KELIM score, personalized oncology

## Abstract

Ovarian cancer remains a leading cause of mortality among gynecological malignancies, with most cases diagnosed at an advanced stage. Treatment approaches include primary cytoreductive surgery followed by chemotherapy or neoadjuvant chemotherapy with interval cytoreduction. This study evaluates the survival outcomes, recurrence patterns, and molecular characteristics of patients undergoing these two treatment strategies. The findings suggest comparable overall survival between the two approaches, though patient selection remains crucial. This study also highlights the role of molecular biomarkers such as BRCA mutations and homologous recombination deficiency (HRD) in predicting responses to treatment. These insights may aid in optimizing personalized therapeutic strategies for ovarian cancer patients.

## 1. Introduction

Ovarian cancer is the most prevalent cause of death for women in developed nations who are diagnosed with gynecological cancers and the second leading fatal cancer of the female reproductive system worldwide, following cervical cancer [1].

Seventy to eighty percent of ovarian cancers are epithelial tumors (EOC), most of them being diagnosed at an advanced stage, where the cancer has spread beyond the ovaries (FIGO—International Federation of Gynaecology and Obstetrics—III–IV), which is associated with a poorer prognosis [2].

For advanced-stage EOC patients, cytoreduction surgery, also known as primary debulking surgery (PDS), followed by an adjuvant treatment, a protocol consisting of chemotherapy (taxane and platinum-based) and anti-VEGF targeted therapy, is currently the gold standard of care [3].

The primary goal of the PDS procedure is to achieve no or less than 10 mm of residual cancer, defined as an optimal or complete cytoreductive surgery [4]. A standard cytoreductive procedure for advanced ovarian cancer generally includes a removal of the ovaries, fallopian tubes, uterus, gastrocolic omentum, as well as enlarged lymph nodes, and a rectosigmoid resection, if required [5,6,7]. A review of 13 studies found that achieving a complete surgical outcome in stage III-IV disease significantly improved the median overall survival by up to 70 months in the patients with optimal cytoreduction, while those with suboptimal outcomes had a median survival of around 30 months [8].

However, as the majority of patients show disease extension over the pelvic brim, a complete cytoreduction is typically unattainable without upper abdominal interventions. This has resulted in the introduction of extensive surgery as a crucial aspect of the ovarian cancer surgical process, which involves at least one of the following procedures alongside the standard operation: a diaphragmatic peritonectomy or partial diaphragmatic resection, an extensive peritonectomy, a splenectomy, a cholecystectomy, pancreatic resection, multiple bowel resections, a gastrectomy, or liver resection. However, extensive surgery is associated with increased morbidity and higher rates of complications due to the complex and invasive nature of the procedures [9,10]. The most common postoperative complications following PDS include pleural effusion, wound infections, ileus (bowel obstruction), and urinary infections [11]. Therefore, severe complications and an extended recovery period may result in a postponement or complete omission of chemotherapy [12,13].

The residual disease following a PDS is the most important prognostic factor regarding the 5-year overall survival (OS); it was found that the patients with smaller metastatic tumors (≤45 mm) at stage IIIC benefit more from primary debulking surgery. This is because smaller metastatic sites are often easier to remove effectively during the initial surgery, improving the survival outcomes [2,14]. Therefore, according to ESMO-ESGO guidelines, it is important that the selection of patients who are appropriate for primary debulking surgery is carried out in a specialized ovarian cancer center as the decision-making process typically involves a multidisciplinary evaluation that weighs the risks and benefits of the procedure based on individual patient characteristics [15]. Poor operative conditions, such as significant medical comorbidities, low performance status, advanced age, or patients with apparently unresectable disease, where it is impossible to achieve an optimal cytoreduction, contraindicate PDS. According to the literature, unresectable disease is often defined as ascites over 1000 mL, omental spread to the spleen greater than 1 cm, liver parenchymal involvement over 1 cm, or disease affecting areas such as the porta hepatis, diaphragm, peritoneum, or suprarenal para-aortic lymph nodes, with each exceeding 1 cm [16,17,18,19,20,21,22].

Systemic therapy consisting of paclitaxel (175 mg/m^2^)—carboplatin (area under the curve (AUC) 5–6), administered every three weeks for six cycles, represents the standard first-line chemotherapy for patients with advanced ovarian cancer. The weekly schedule with paclitaxel (60 mg/m^2^) and carboplatin (AUC 2) may serve as an alternative for frail patients [1]. Also, several randomized controlled trials (RCTs) investigated comparatively the effectiveness of chemotherapy with and without bevacizumab, an anti-VEGF antibody [23]. The majority of these studies (GOG2018, OCEANS, AURELIA, and GOG 2013) demonstrated a significantly improved progression-free survival in the patients with stage III-IV EOC receiving the antibody in addition to paclitaxel—carboplatin therapy [24,25,26,27,28].

Neoadjuvant chemotherapy (NACT) was proposed as an alternative to address the shortcomings of primary surgery in patients with advanced ovarian cancer. Only about 20–25% of the patients attained total remission with traditional primary debulking surgery and adjuvant therapy, with most experiencing a recurrence of the disease within five years [3]. Moreover, achieving a complete cytoreduction (the absence of residual disease) is critical in improving a prognosis; however, it can be difficult to acquire in advanced cases. Finally, the high risk of postoperative complications with PDS led to the consideration of NACT as a way to reduce the tumor burden before surgery, potentially decreasing surgical risks and improving patient outcomes [13,29].

Numerous studies have indicated that NACT followed by interval debulking surgery (IDS), as an alternative treatment strategy in advanced epithelial ovarian cancer (expecting a non-optimal cytoreduction during PDS), shows non-inferior survival outcomes and is associated with less postoperative morbidity when compared with PDS. Multiple randomized, controlled trials (RCTs, e.g., the EORTC 55,971 trial, the CHemotherapy OR Upfront Surgery—CHORUS trial, and the JCOG 0602 trial) have indicated that patients who had undergone NACT-IDS experienced significantly fewer side effects and lower surgical mortality rates compared to those who underwent PDS. Survival outcomes, including progression-free survival (PFS) and overall survival (OS), were similar between NACT-IDS and PDS [30,31,32,33].

One of the first RCTs, conducted by Vergote et al. in 2010, involving 670 patients, found that 80.6% of the patients achieved a residual tumor maximum size of 1 cm or less after IDS, compared to only 41.6% following PDS. Also, the number of postoperative adverse effects and the mortality rates were higher in the primary debulking group, although the hazard ratios for death and disease progression were similar across both treatment groups. The investigators determined that NACT followed by IDS was not inferior to primary debulking followed by chemotherapy, and that a complete resection happened more frequently in the NACT cohort [31]. Other subsequent RCTs published in 2011 and 2015 also supported these findings; in a multicenter randomized controlled trial conducted by Kehoe et al. involving 552 women, the median overall survival rates were similar between the two groups, reporting 22.6 months for the primary surgery cohort and 24.1 months for the primary chemotherapy cohort. The hazard ratio for mortality favored primary chemotherapy at 0.87, with a one-sided 90% confidence interval with an upper bound of 0.98. Moreover, the rates for grade 3 or 4 postoperative complications and mortality within 28 months were significantly higher in the primary surgery group (24% compared to 14% and 6% compared to less than 1%, respectively). The authors determined that for women with stage III or IV ovarian cancer, primary chemotherapy is not inferior to primary surgery, thus recognizing it as an appropriate standard of care for this demographic [32].

However, despite multiple clinical trials, the clinical implications of NACT still remained debatable. The absence of agreement on NACT was evident in the surveys conducted within gynecologic society groups: 70% of the members of the European Society of Gynecological Oncology (ESGO) believed the evidence was adequate to recommend NACT, while in the population predominantly from the US, 82% of the Society of Gynecologic Oncologists (SGOs) members claimed there was insufficient evidence to support the use of neoadjuvant chemotherapy in epithelial ovarian cancer (EOC) according to a 2010 survey. In the follow-up poll conducted five years later, 68% of SGO members still considered the data insufficient to support the routine use of NACT [34].

NACT provides several clinical advantages over PDS in the treatment of advanced ovarian cancer, as evidenced by the comparative data from clinical studies. These benefits include lower rates of serious postoperative adverse effects in the NACT group (6%) versus the PDS group (29%), a decreased need for stoma creation (5.9% in the NACT group compared to 20.4% in the PDS group), reduced rates of bowel resection (often necessary when tumors spread extensively) among NACT patients (13.0%) compared to those undergoing PDS (26.6%), and reduced immediate postoperative mortality rates (0.6% for NACT compared to 3.6% for PDS) [35]. NACT indicates clinical benefits in reducing the immediate surgical impact. Studies such as the SCORPION trial suggest a potential improvement in quality of life with neoadjuvant chemotherapy; however, the quality of life (QoL) assessments, particularly using the EORTC QoL scale, remained imprecise and inconclusive [35,36].

Patient selection for neoadjuvant chemotherapy, in accordance with ESMO 2024 criteria, should be based on the likelihood of achieving a complete cytoreduction [1]. ESMO-ESGO indicates patients with conditions like a diffuse deep infiltration of the root of the small bowel mesentery, stomach, or duodenum, as well as the head or middle part of the pancreas; an involvement of the coeliac trunk, hepatic arteries, or left gastric artery; diffuse carcinomatosis of the small bowel involving extensive regions in such a way that resection would result in short bowel syndrome (remaining bowel < 1.5); central or multisegmental parenchymal liver metastases; multiple parenchymal lung metastases; nonresectable lymph nodes; and brain metastases are unsuitable for primary debulking surgery [37]. In such cases where complete cytoreductive surgery is not feasible, it is essential to obtain adequate biopsy tissue for a histological and molecular analysis. For these patients, it is recommended to administer three cycles of neoadjuvant chemotherapy, followed by interval cytoreductive surgery and three cycles of paclitaxel—carboplatin. bevacizumab may be considered in the neoadjuvant setting prior to ICS. In situations in which ICS is not possible and there is no evident disease progression, it is advisable to administer three additional cycles of paclitaxel—carboplatin, either alone or in combination with bevacizumab [1].

## 2. Materials and Methods

This study aims to provide a “real-life” presentation of the outcomes for patients with stage III-IV high-grade serous ovarian cancer who received neoadjuvant chemotherapy followed by interval cytoreductive surgery compared with primary cytoreductive surgery and adjuvant chemotherapy. This is a topic of active debate within the scientific community. It also compares outcomes between the two treatment strategies, investigates the risk factors for recurrence, and assesses the impact of BRCA mutations and HRD deficiency on patient outcomes. To benefit the 5-year overall survival outcomes, we documented data from the patients who underwent primary surgery or a first cycle of neoadjuvant chemotherapy between 1 January 2018 and 31 December 2018, at a Cancer Institute in Bucharest, Romania, i.e., the Bucharest Institute of Oncology with “Prof. Dr. Al. Trestioreanu”.



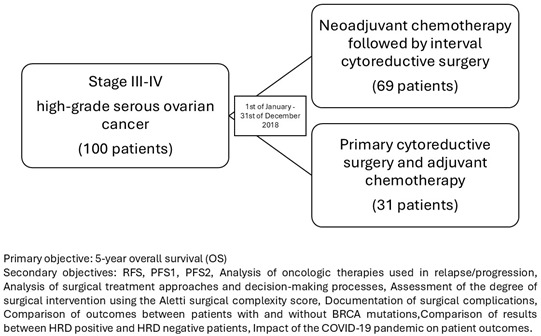



### 2.1. Selection Criteria

The inclusion criteria were as follows:Patients over 18 years of age.An ECOG performance status of 0–1 at the time of surgery.Radiologically suspected or histologically confirmed and newly diagnosed primary epithelial ovarian cancer, fallopian tube carcinoma, or peritoneal carcinoma (FIGO stage IIIA-IVB).Patients who underwent primary surgery or a first course of neoadjuvant chemotherapy between 1 January 2018 and 31 December 2018.Surgery performed by laparotomy with a maximal effort trial.Adequate bone marrow function: absolute neutrophil count (ANC) ≥ 1.5 × 10^9^/L.Preoperative imaging that ruled out unresectable disease according to ESGO criteria.The presence of relapses and/or relapses documented in the observation sheets.

The exclusion criteria were as follows:Non-epithelial malignant ovarian neoplasms and borderline tumors.Other neoplasms that were diagnosed in the last 5 years.Recurrent ovarian cancer.Unresectable parenchymal lung metastases, liver metastases, or bulky lymph node metastases in the mediastinum, as seen on preoperative imaging.Previous chemotherapy for ovarian cancer or abdominal/pelvic radiotherapy.Pregnant women at the time of diagnosis.

This study aims to obtain data from approximately 100 patients who underwent cytoreductive surgery in 2018. By assessing these factors, this study aims to provide robust real-world evidence to guide clinical decision-making in the treatment of advanced ovarian cancer.

### 2.2. Study Design

This is an observational, retrospective study on a cohort of ovarian cancer patients treated since 2018 at the Institute of Oncology by “Prof. Dr. Dr. Al. Trestioreanu” in Bucharest, Romania. This study includes the patients who fulfilled the specific inclusion and exclusion criteria previously presented.

The primary objective is as follows:

The primary objective is to compare the 5-year overall survival (OS) rates between the patients who underwent primary cytoreductive surgery and those who received neoadjuvant chemotherapy followed by interval cytoreductive surgery for stage IIIA-IVB ovarian cancer.

The secondary objectives are as follows:Relapse-free survival (RFS).Progression-free survival 1 (PFS1) or Progression-free survival 2 (PFS2).An analysis of the oncologic therapies used in relapse/progression.An analysis of the surgical treatment approaches and decision-making processes for the patients with advanced epithelial ovarian cancer.An assessment of the degree of surgical intervention needed, using the Aletti surgical complexity score.Documentation of surgical complications.A comparison of outcomes between the patients with and without BRCA mutations.A comparison of the results between HRD-positive and HRD-negative patients.The impact of the COVID-19 pandemic on patient outcomes.

### 2.3. Definitions

Overall survival (OS): the time from first treatment (chemotherapy for neoadjuvant cases and surgery for primary cytoreductive cases) to death from ovarian cancer or the last evaluation.

Relapse free survival (RFS): the time from the time of the initial cancer treatment until the disease recurs (relapses) or death from any cause occurs.

Progression Free Survival 1 (PFS1): the time interval from the start of line I treatment until disease progression or death, or the date of the last evaluation. Disease progression is defined as clinically, biologically, or imaging-detected tumor growth or death without previously documented tumor progression.

Progression free survival 2 (PFS2): the time interval between disease progression after first-line treatment and disease progression after second-line treatment or death from any cause. Maintenance treatments after chemotherapy are not considered separate lines of treatment.

### 2.4. Variables

This study included several variables to assess patient demographics, tumor characteristics, treatment modalities, and outcomes.

The patients were divided into the following age groups: 40–49 years, 50–59 years, 60–69 years, and over 70 years.

The body mass index (BMI) categories were defined as under 18.5 (underweight), 18.5–24.9 (normal weight), 25.0–29.9 (overweight), and over 30.0 (obese). The tumor histology was classified as high-grade serous, endometrioid, and mixed histology. The locations of the tumors were documented as the ovaries, fallopian tubes, or peritoneum. The patients were staged according to the FIGO system: IIIA, IIIB, IIIC, IVA, or IVB, as mentioned in the inclusion criteria. Performance status was assessed using the ECOG scale, and patients with ECOG 0 or 1 were included.

The BRCA mutation test results were categorized as follows: BRCA1 positive, BRCA2 positive, no mutation (BRCA wild-type), or the test was not performed. Homologous recombination deficiency (HRD) was recorded as follows: HRD positive, HRD negative, or the test was not performed. The type of surgery performed was categorized into the following: primary cytoreductive surgery or interval cytoreductive surgery. The indication for primary chemotherapy was documented as follows: neoadjuvant chemotherapy or adjuvant chemotherapy. Maintenance treatment after the initial therapy or first recurrence included bevacizumab or Olaparib.

The types of chemotherapy regimens used were documented, and the chemotherapy regimens for recurrent disease were categorized according to platinum sensitivity (platinum-sensitive patients or platinum-resistant patients).

Surgical complexity was assessed using a modified Aletti score with points assigned as follows (Total score: 20 points) [29]:Total hysterectomy or unilateral/bilateral oophorectomy (1 point).Omentectomy (1 point).Pelvic lymphadenectomy (1 point).Para-aortic lymphadenectomy (1 point).Small bowel resection (1 point).Large bowel resection (2 points).Large bowel resection with colo-colic anastomosis (2 points).Large bowel resection with colo-rectal or colo-anal anastomosis (3 points).Splenectomy (2 points).Diaphragm resection/decortication (2 points).Urological shunt (2 points).Liver resection (2 points).

Surgical complexity was then categorized into ≤ 3 points (low complexity), 4–7 points (intermediate complexity), and ≥8 points (high complexity).

Surgical complications have been documented and categorized.

CA-125 levels were assessed during the first three cycles of neoadjuvant or adjuvant chemotherapy and used to calculate the KELIM scores. [38]

The influence of the COVID-19 pandemic on patient monitoring and treatment was also assessed.

## 3. Results

### 3.1. Clinicopathological and Genetic Factors

In this study, the distribution of patients was based on ECOG performance score, age, and body mass index (BMI). The majority of patients (84%) had an ECOG score of 0, indicating a good general condition with no functional limitations. In terms of age, the majority (66%) were in the 60–69 age range. The BMI distribution showed that 67% of the patients were overweight (BMI between 25 and 29.9) and 14% were obese, reflecting a high prevalence of overweight in this sample.

The distribution of patients according to FIGO stage and histologic type reveals that the majority were diagnosed with advanced disease. Thus, 36% of the patients were in stage IVA and 19% were in stages IIIB and IVB. Histologically, 91% of the patients had high-grade serous carcinoma, the most common form of ovarian cancer. In terms of tumor localization, the majority of patients (95%) had tumors localized to the ovaries, indicating a major prevalence of ovarian involvement.

The distribution of BRCA mutations and homologous recombination deficiency (HRD) status reflect the genetic profile of the patients. The BRCA test was not performed in 9% of the patients, and 67% had no BRCA mutations, while 16% had BRCA1 mutations and 8% had BRCA2 mutations. Regarding HRD, the test was not performed in 65% of the patients, and of those tested, 11% were HRD positive, which may influence the treatment options with PARP inhibitors.

The variables were analyzed in each of the two groups and the corresponding results can be seen in Table 1.

Most of the patients in both groups were aged 60–69 years. There were no significant differences between the two groups (*p* = 0.8633). Most of the patients in both groups had a BMI between 24.9 and 29.9. There were no significant differences between the groups (*p* = 0.9795). The vast majority of patients (91%) had high-grade serous cancer, with little variation between the groups (*p* = 0.1796). Most tumors were located in an ovary (95%), with no significant differences between the groups. Patients in the neoadjuvant chemotherapy group had a balanced distribution between stages III and IV, while the majority of the patients in the adjuvant group were stage IV (*p* < 0.001). Among the patients classified as FIGO stage IVB (n = 19) in the adjuvant group, the most frequent sites of metastatic disease were the supradiaphragmatic lymph nodes, observed in eight patients (42.1%), followed by pleural effusion with positive cytology in six patients (31.6%). Parenchymal liver metastases were reported in three patients (15.8%), and pulmonary nodules suspected of metastasis were present in two patients (10.5%). Within the neoadjuvant chemotherapy group, two patients were also classified as FIGO stage IVB. One patient presented with isolated supraclavicular lymph node metastasis, while the other had pleural effusion with positive cytology.

Approximately 84% of the patients in both groups had an ECOG score of 0 (no significant limitations in the activities of daily living) (*p* = 0.9863). In both groups, the majority of patients had no BRCA mutations. BRCA tests were not performed in a small proportion of patients in both groups (*p* = 0.1602). Positive HRD was significantly more frequent in the neoadjuvant chemotherapy group (*p* < 0.001), but the low number of tests precludes the conclusion of a significant result.

### 3.2. Analysis of Surgical Interventions

Figure 1 is a density plot comparing the distribution of modified Aletti scores for two groups of patients: those who received adjuvant chemotherapy and those who received neoadjuvant chemotherapy.

The horizontal axis (X) represents the modified Aletti score, which is a score that assesses the complexity of surgery in ovarian cancer patients. A higher score indicates a more complex surgical procedure. The vertical axis (Y) represents the relative density of the distribution of scores in each group. The higher the density at a given point on the *X*-axis, the more cases are concentrated around that score.

The adjuvant group (red) represents the patients who received adjuvant chemotherapy after primary cytoreductive surgery. The neoadjuvant group (blue) represents the patients who received neoadjuvant chemotherapy before interval cytoreductive surgery.

The neoadjuvant group has a distribution of modified Aletti scores more clustered around low values (approximately 4–6), indicating that, in this group, the surgeries were generally less complex. The density curve shows a clear peak in the 4–5 range, suggesting that most patients had low surgical complexity scores.

The distribution of the adjuvant group is shifted towards higher Aletti scores, suggesting a higher complexity of surgeries. The density curve for this group shows a peak around 8–10, indicating more complex surgeries compared to the neoadjuvant group.

The patients operated on after neoadjuvant chemotherapy tended to have lower Aletti scores, indicating that the surgeries were less complex in general, probably due to the effect of chemotherapy on reducing tumor volume before surgery. The patients operated per-primam showed higher scores, suggesting more complex surgery.

Surgical complications occurred in 28.99% of the patients who received neoadjuvant chemotherapy followed by interval cytoreduction and in 51.61% of those who received primary cytoreduction.

### 3.3. Analysis of Treatments Performed

The initial chemotherapy treatment (neoadjuvant or adjuvant) for all patients used the standard combination of paclitaxel and carboplatin. In the neoadjuvant chemotherapy group, 65.22% of the patients (46) had a favorable KELIM score, while in the adjuvant chemotherapy group, it was found in 45.16% of the patients (14) (Figure 2 and Figure 3).

After adjuvant treatment, 47 patients (47.00%) received maintenance treatment with bevacizumab. At that time in Romania, Olaparib was not reimbursed as a maintenance treatment after a first-line treatment, but only as a maintenance treatment after a first relapse. The combination of Olaparib plus bevacizumab was not available either. For this reason, patients only received bevacizumab as a maintenance treatment after first-line treatment

After the first relapse, 24 patients (24.00%) received Olaparib maintenance treatments and 46 (46.00%) received bevacizumab.

Depending on the time to first recurrence, the patients were considered platinum-resistant (17.00%) or platinum-sensitive (83.00%). This was taken into account in determining subsequent treatments.

Table 2 details the chemotherapeutic regimens used to treat recurrences and the subsequent lines of treatment for ovarian cancer patients.

In first-line treatment (100 patients), carboplatin + paclitaxel is the most commonly used regimen (74% of patients) and is the standard treatment for platinum-sensitive tumors. Other less common regimens included carboplatin + gemcitabine (9%), topotecan (8%), and liposomal pegylated doxorubicin (7%).

In second-line treatment (48 patients), carboplatin + paclitaxel remains the dominant treatment as a percentage (54.17% of cases), but with a decrease compared to the first-line treatment. Gemcitabine and topotecan are also used (16.67% and 14.58%, respectively).

In the third line of treatment (20 patients), gemcitabine was the most commonly used regimen (40%), followed by liposomal pegylated doxorubicin (20%) and topotecan (15%).

In the fourth line of treatment (eight patients), the options vary, but paclitaxel is the most commonly used (37.5%), followed by etoposide and gemcitabine (25% each).

### 3.4. Survival Outcomes

The mean RFS (mean time to first relapse) for the neoadjuvant chemotherapy group (N1) is 13.75 months, compared with 11.38 months for the adjuvant chemotherapy group (N2).

The median RFS (time to relapse for 50% of patients) is identical in both groups: 12 months.

The patients in the neoadjuvant group have a slightly longer relapse-free survival, but the medians are the same for both groups, suggesting that the treatments offer similar benefits.

The mean PFS1 for the neoadjuvant group (N1) is 14.40 months, compared with 13.83 months for the adjuvant group (N2).

The median PFS1 is slightly lower for the neoadjuvant group (11 months) compared to the adjuvant group (12 months).

Although the mean differences are minimal, the adjuvant-treated group has a slightly longer median PFS1 time, indicating a marginal benefit in this phase.

The mean PFS2 for the neoadjuvant group is 6.92 months compared with 6.23 months for the adjuvant group.

The median PFS2 is almost identical in both groups: 7 months (neoadjuvant) and 6 months (adjuvant).

There are no major differences between groups in progression-free survival after second-line treatment.

The mean PFS3 is 3.54 months for the neoadjuvant group compared to 3.88 months for the adjuvant group. The median PFS3s are identical: 3 months in both groups.

Both groups have similar survival rates after the third line of treatment, with little variation in the mean value.

The mean PFS4 is 2.5 months for the neoadjuvant group compared to 4 months for the adjuvant group. The median PFS4s are identical to the means: 2.5 months (neoadjuvant) and 4 months (adjuvant).

The group treated with adjuvant chemotherapy appears to have a longer progression-free survival during the fourth line of treatment, but the number of patients is very small (only two for the neoadjuvant and six for the adjuvant).

The mean OS is similar between the groups: 31.78 months for neoadjuvant and 30.74 months for adjuvant. The median OS is slightly higher in the neoadjuvant group (27 months) than in the adjuvant group (26 months). Overall survival is very similar between the two groups, with no significant differences.

The differences between neoadjuvant and adjuvant chemotherapy are minimal in terms of relapse-free survival (RFS), progression-free survival (PFS), and overall survival (OS). However, in some treatment lines, neoadjuvant chemotherapy appears to offer a slight advantage, particularly for RFS, PFS1, PFS2, and OS. The small number of patients in some phases of this study limits the statistical significance of these conclusions.

The patients treated with neoadjuvant chemotherapy appear to show slightly better survival rates than those operated per-primam who received adjuvant chemotherapy at certain time points, but there is not a very large difference between the two curves (Figure 4).

The *p*-value (*p* = 0.08) indicates that the difference between the two groups is not considered statistically significant at a common threshold of significance (*p* < 0.05). However, the *p* is relatively close to 0.05, which could suggest a trend towards a clinically relevant difference, but more data may be needed to confirm this.

Although the group treated with neoadjuvant chemotherapy shows a slightly longer relapse-free survival than the group treated with adjuvant chemotherapy, the difference is not large enough to be considered statistically significant.

In terms of PFS1, in the patients who received either adjuvant chemotherapy (black line) or neoadjuvant chemotherapy (red line), the two survival curves are very similar, indicating that there is no significant difference in progression-free survival (PFS) between the two groups (Appendix A).

The *p*-value of 0.745 indicates that there is no statistically significant difference in survival between the two groups, so both adjuvant and neoadjuvant chemotherapy had comparable effects on PFS1.

The same is true for PFS2, with a *p*-value of 0.525 indicating no statistically significant difference between the two groups in PFS2 (Appendix A).

Also, PFS3 is similar between the two groups, with *p* statistically insignificant (*p* = 0.887) (Appendix A).

The previously described numerical difference in OS in the two groups is not statistically supported. A *p*-value of 0.579 indicates that there is no statistically significant difference in overall survival between the patients in the two groups (Appendix A).

Patients who received Olaparib as a maintenance therapy at first relapse had a significantly longer overall survival compared to those who received bevacizumab, and this difference is highly statistically significant (*p* < 0.001). Olaparib appears to be a more effective maintenance treatment in improving overall survival in this setting (Figure 5).

Although the results suggest a significant difference between the groups treated with bevacizumab and Olaparib, it is important to note that the sample size was relatively small (46 patients treated with bevacizumab and 24 with Olaparib). Thus, these findings should be interpreted with caution, and future studies with a larger sample size are needed to confirm these findings and to further evaluate the efficacy of the treatments.

### 3.5. Influence of Pandemic COVID-19

The COVID-19 pandemic had multiple documented influences on the treatment of cancer patients. In the group of ovarian cancer patients analyzed, 87% were not affected by the pandemic in terms of treatment (Appendix A).

However, 7% had their treatments delayed due to the pandemic and 6% of the patients died, possibly as a result of complications associated with a SARS-CoV-2 infection. Although delays did occur, these results show that most patients were able to continue therapy without major interruptions (Appendix A).

## 4. Discussion

The results of this retrospective study make a real-world contribution to the general debate on the optimal management of patients with advanced-stage high-grade serous ovarian cancer (FIGO III-IV). Although both primary cytoreduction followed by adjuvant chemotherapy and neoadjuvant chemotherapy followed by interval cytoreduction have shown similar outcomes in progression-free survival and overall survival, the choice between these two approaches remains a clinical challenge.

Although several randomized controlled trials (RCTs) have compared primary cytoreductive surgery with neoadjuvant chemotherapy followed by interval debulking surgery, our study provides additional insights into the real-world application of these treatment strategies within the context of a Romanian tertiary oncology center. The retrospective design allowed us to capture the variability in clinical decision-making influenced by factors such as access to specialized gynecologic oncology surgeons, institutional delays in surgical scheduling, and the limited availability of molecular testing or maintenance therapies like PARP inhibitors. These elements often fall outside the scope of controlled clinical trials but significantly shape treatment outcomes in everyday practice. Therefore, our findings offer valuable information for healthcare systems facing similar constraints and highlight the need for context-adapted treatment pathways in advanced ovarian cancer management.

In terms of overall survival (OS) and progression-free survival (PFS), the results suggest that there are no significant differences between the two therapeutic approaches, with both showing comparable survival rates. This observation is consistent with previous clinical trials, which have indicated that both neoadjuvant and adjuvant chemotherapies are viable options for the treatment of patients with advanced ovarian cancer.

The patients who received neoadjuvant chemotherapy generally had a less complex surgery (lower Aletti score). This trend may be explained by tumor volume reduction before surgery facilitating complete cytoreduction. However, it should be noted that the patients who underwent primary cytoreduction required a more complex surgery, which may influence the risk of postoperative complications.

Although no major between-group differences in BRCA status were observed, HRD- positive patients tended to respond better to PARP inhibitor-based treatments, suggesting that this biomarker may play an important role in personalizing treatment for these patients.

A small percentage of patients were directly affected by delays or complications caused by the pandemic, which indicates adequate management of oncology patients during this difficult period. However, the psychological impact of the pandemic and fears of treatment interruptions were felt by a proportion of patients, emphasizing the importance of ensuring continuity of care even in a health crisis.

One of the main limitations of this study is its retrospective nature, and the relatively small number of patients included, which may limit the statistical power of the conclusions. In addition, the treatments and access to resources varied according to the pandemic context, which may have influenced the results. Also, over the past 5 years, therapeutic advances and drug approvals/refunds have generated other potential types of therapeutic conduct for clinical cases similar to those in this study. The absence of PARP inhibitors such as Olaparib as maintenance therapies during the study period represents a significant limitation that may have influenced survival outcomes, particularly in the patients with BRCA mutations or an HRD-positive status. Therefore, the results reported here may underestimate the therapeutic potential achievable with the currently available targeted therapies in a modern clinical setting.

The exploratory subgroup analysis comparing the outcomes following bevacizumab versus Olaparib treatment at first relapse was based on a limited number of patients and should be interpreted with caution. Larger prospective studies are needed to validate these findings and provide further evidence in support of one of the two therapeutic approaches.

Another notable limitation of our study is the absence of consistent documentation regarding the completeness of cytoreductions (R0 status) in surgical and histopathological reports from 2018. As a result, we were unable to assess the potential impact of a complete cytoreduction on survival outcomes.

In the absence of a national cancer registry or a dedicated institutional database for ovarian cancer, the inclusion of patients was based on a manual review of the medical records archived in the hospital. We selected consecutive cases of newly diagnosed advanced high-grade serous ovarian cancer until reaching a target sample size of 100 patients. While this approach was necessary due to infrastructure constraints, it may have introduced selection bias and cohort heterogeneity, limiting the generalizability of the findings.

The results of this study suggest that both primary cytoreduction and neoadjuvant chemotherapy followed by surgery are viable options for patients with advanced ovarian cancer. However, the choice between neoadjuvant chemotherapy followed by interval cytoreduction and primary cytoreduction followed by adjuvant chemotherapy for the treatment of advanced-stage high-grade serous ovarian cancer depends on a number of patient-specific clinical and prognostic factors.

The factors influencing the decision include the following:In cases where the tumor is very extensive, which would make a complete cytoreduction technically difficult or impossible without significant risk to the patient, neoadjuvant chemotherapy is preferred to reduce tumor size and facilitate a safer and more effective interval cytoreduction.Patients who have poorer overall health (e.g., ECOG status ≥ 1) and who would have a harder time tolerating major surgery at diagnosis may benefit from a neoadjuvant chemotherapy approach. This gives them time to improve their general condition before surgery.One of the most important factors in successful treatment is achieving a complete cytoreduction (no visible residual disease). If, at a preoperative evaluation, the medical team feels that a complete cytoreduction is not achievable, neoadjuvant chemotherapy may be used to shrink the tumors before surgery.Some medical centers do not have the resources or advanced surgical expertise to perform complex cytoreductive surgery. In these cases, neoadjuvant chemotherapy may be a safer and more effective option to prepare the patient for surgery.Patients with BRCA mutations or homologous recombination deficiency (HRD) may have a better response to PARP inhibitor-based therapies, which are more effective after neoadjuvant chemotherapy, thus providing an additional advantage for this approach.Primary cytoreduction is usually preferred for patients in a good general condition (ECOG 0-1), with surgically resectable disease, and at a lower risk of complications. It is associated with a better prognosis, especially in cases where complete cytoreduction can be achieved.Neoadjuvant chemotherapy is preferred when surgery at the outset is risky or impossible because of tumor enlargement or when the patient has comorbidities that make major surgery risky. This approach is also recommended when the medical team estimates that the chances of achieving a complete cure are low.

The data from the current study are consistent with the data in the literature, including the studies mentioned in the introduction and meta-analyses on this topic [24,25,26,27,28,29,30,31,32,33].

One of the meta-analyses (Cochrane) used pooled data from four studies, and these showed no significant differences in OS (HR 0.96, 95% CI 0.86–1.08) or PFS (HR 0.98, 95% CI 0.88–1.08). However, neoadjuvant chemotherapy significantly reduced major postoperative adverse events (6% in the neoadjuvant chemotherapy group vs. 29% in the per-primam surgical intervention group), the need for stoma fitting (5.9% vs. 20.4%), and the risk of bowel resection during surgery (13% vs. 26.6%). Postoperative mortality was also lower in the neoadjuvant chemotherapy group (0.6% vs. 3.6%). Quality of life outcomes were variable and imprecise [35].

A study conducted in China on 110 patients undergoing primary cytoreduction, and 51 patients receiving neoadjuvant chemotherapy followed by interval cytoreduction surgery showed that the former group had a longer PFS compared to the latter (*p* = 0.029). The R0 resection rate was 57.8%. All but one patient received platinum-based chemotherapy, and 65.2% were platinum-sensitive. A subgroup analysis indicated that the patients treated with neoadjuvant chemotherapy and with residual disease had the lowest PFS (*p* = 0.001). A multivariate analysis confirmed that treatment with neoadjuvant chemotherapy with residual disease was an independent factor associated with poorer PFS (*p* = 0.04). However, neoadjuvant chemotherapy did not influence OS in the univariate or multivariate analyses [39].

In an article entitled “Knowledge, attitudes and practices related to the COVID-19 pandemic among Romanian adults with cancer: a national cross-sectional study” (Authors: A.S. Gheorghe, Ș. M. Negru, C. Nițipir, L. Mazilu, M. Marinca, B. Gafton, T. E. Ciuleanu, M. Schenker, R. D. Dragomir, A. D. Gheorghe, P. O. Stovicek, M. Bandi-Vasilica, A. C. Boț, R. I. Mihăilă, D. L. Zob, A. L. Kajanto & D. L. Stănculeanu), published in *ESMO Open*, vol. 6, January 2021, we analyzed the influence of the pandemic on oncology patients [40].

The COVID-19 pandemic has imposed major challenges for oncology care providers because of the special precautions required for cancer patients, whose immune systems are often compromised. This study aimed to describe the level of knowledge, attitudes and practices related to COVID-19 among Romanian cancer patients in order to assess the impact of the pandemic and the effectiveness of response measures.

The research was conducted through a multicenter cross-sectional study involving 1585 oncology patients from seven hospitals in Romania. The participants completed a questionnaire with 64 questions on knowledge, attitudes, and practices related to COVID-19 between April and May 2020.

By separately analyzing the patients with primary ovarian or peritoneal cancer (82 out of 1585), we obtained information on risk perceptions, fears, and the influence of COVID-19 on disease progression in this diagnostic category (Appendix A).

A total of 67.50% of the patients (n = 54) considered their oncologic diagnosis as putting them at an additional risk compared to the rest of the population to contract the new coronavirus. Conversely, 88.61% (n = 70) did not think that this risk justified delaying treatment. Almost a third, 35.00% (n = 28), were more afraid of oncologic disease progression and only 7.50% (n = 6) were more afraid of coronavirus infection.

## 5. Conclusions

In conclusion, the choice between these two approaches has to be individualized, taking into account tumor characteristics as well as the patient’s general condition and available resources. The decision is based on a balance between the risks associated with each strategy and the long-term benefits for the patient.

The patients who underwent NACT-IDS demonstrated lower rates of perioperative complications and surgical morbidity, suggesting an advantage for this approach in cases where a complete cytoreduction is initially deemed unachievable. However, the success of either strategy depends on a careful preoperative assessment and patient selection, particularly with respect to tumor burden and resectability.

Molecular biomarkers, including BRCA status and HRD, emerged as critical factors in predicting responses to treatment. HRD-positive patients showed better responses to PARP inhibitors, highlighting the need for genetic profiling to guide treatment decisions. Additionally, KELIM score proved to be a useful predictor of chemotherapy responses, offering further potential for personalized treatment planning.

Considering our findings and the practical challenges observed in routine clinical care, we believe that future prospective studies should explore personalized treatment strategies guided by molecular biomarkers such as HRD status, as well as dynamic response predictors like a KELIM score. Integrating these tools into therapeutic decision-making may improve the outcomes for patients with advanced ovarian cancer and support the evolution toward more individualized, biomarker-driven treatment algorithms.

Although this study was limited by its retrospective nature and single-center design, its findings align with the existing literature and reinforce the importance of individualized treatment strategies for ovarian cancer. Future prospective studies incorporating larger patient cohorts and novel therapeutic agents will be essential to refine treatment algorithms and further enhance patient outcomes.

Ultimately, the integration of clinical, surgical, and molecular factors will enable more precise and effective treatment approaches, improving the prognosis and quality of life for patients with advanced ovarian cancer.

## Figures and Tables

**Figure 1 cancers-17-01314-f001:**
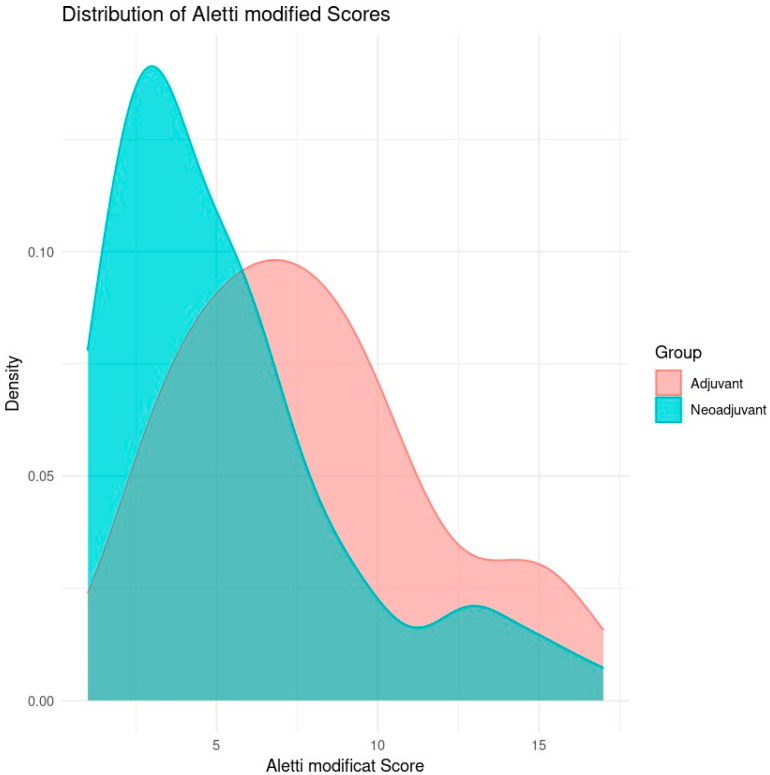
Distribution of Aletti score by type of surgery (Aletti X sq *p*-value = 0.1627).

**Figure 2 cancers-17-01314-f002:**
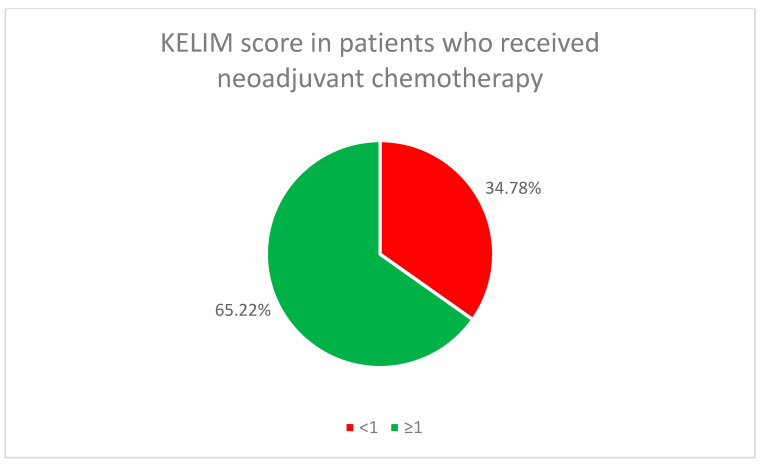
KELIM score in patients who received neoadjuvant chemotherapy.

**Figure 3 cancers-17-01314-f003:**
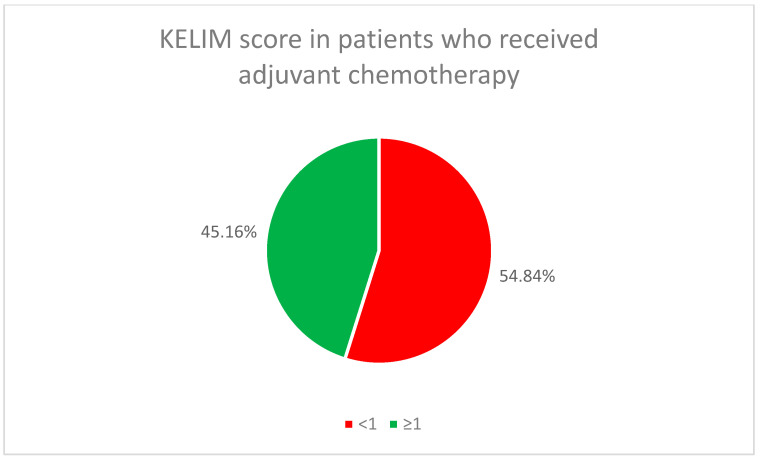
KELIM score in patients who received adjuvant chemotherapy.

**Figure 4 cancers-17-01314-f004:**
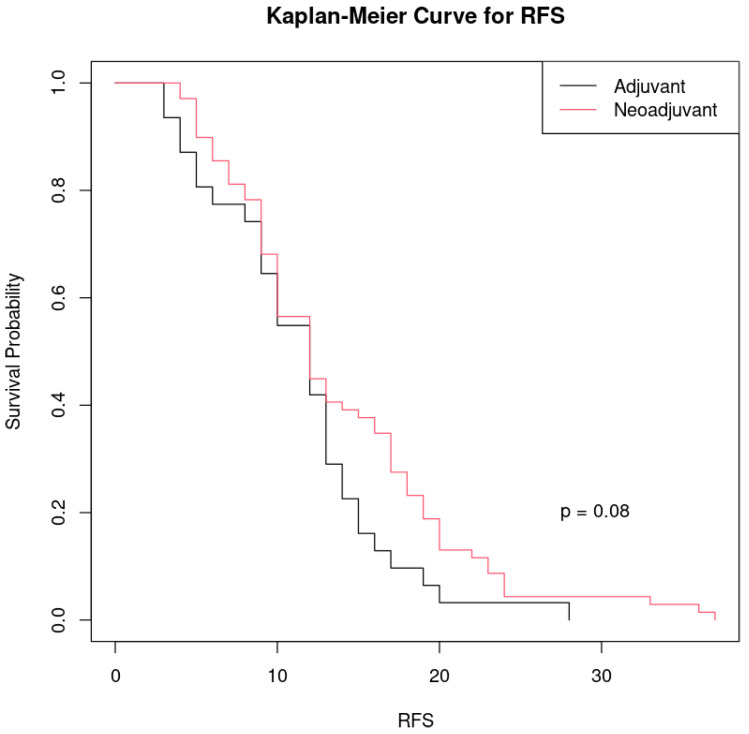
Kaplan–Meier curve comparing recurrence-free survival (RFS) between the two groups of patients treated with different therapeutic approaches to ovarian cancer: black line: patients who received adjuvant chemotherapy after primary cytoreduction and red line: patients who received neoadjuvant chemotherapy before surgery.

**Figure 5 cancers-17-01314-f005:**
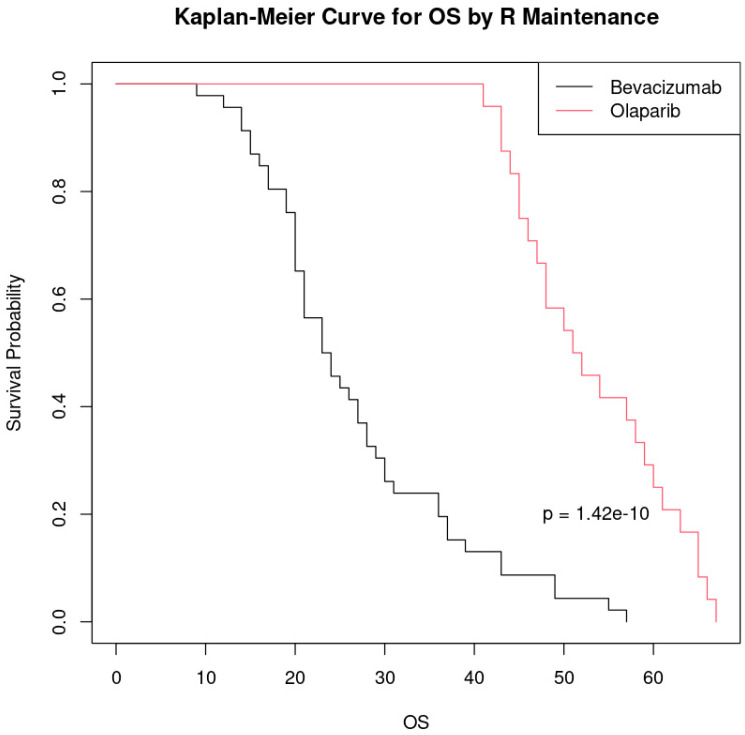
Kaplan–Meier curve comparing overall survival (OS) by maintenance treatment at first relapse.

**Table 1 cancers-17-01314-t001:** Patient and ovarian cancer characteristics in the two groups, primary cytoreductive surgery plus adjuvant chemotherapy and neoadjuvant chemotherapy plus interval cytoreduction (*p*—Fischer exact test).

Variable	Patients N = 100	Neoadjuvant Chemotherapy N1 = 69 (69%N)	Adjuvant Chemotherapy N2 = 31 (31%N)	*p*
**Age**				
40–49	9	3 (9.68)	6 (8.7)	0.8633
50–59	18	7 (22.6)	11 (15.9)	
60–69	66	19 (61.3)	47 (68.1)	
>70	7	2 (6.45)	5 (7.25)	
**BMI**				
<18.5	5	5 (16.1)	9 (13.0)	0.9795
18.5–24.9	14	21 (67.7)	46 (66.7)	
24.9–29.9	67	4 (12.9)	10 (14.5)	
>30	14	1 (3.23)	4 (5.8)	
**Histology**				
High-grade serous carcinoma	5	3 (9.68)	2 (2.9)	0.1796
Endometroid carcinoma	91	28 (90.3)	63 (91.3)	
Mixed	4	-	4 (5.8)	
**Location**				
Fallopian tube	3	1 (3.23)	2 (2.9)	-
Ovary	95	30 (96.8)	65 (94.2)	
Peritoneum	2	-	2 (2.9)	
**FIGO Stage**				
IIIA	8	8 (25.8)	-	<0.001
IIIB	19	10 (32.3)	9 (13.0)	
IIIC	18	9 (29.0)	9 (13.0)	
IVA	36	2 (6.45)	34 (49.3)	
IVB	19	2 (6.45)	17 (24.6)	
**ECOG**				
0	84	26 (83.9)	58 (84.1)	0.9863
1	16	5 (16.1)	11 (15.9)	
**BRCA**				
BRCA 1+	16	5 (16.1)	11 (15.9)	0.1602
BRCA 2+	8	2 (6.45)	6 (8.7)	
No mutation	67	24 (77.4)	43 (62.3)	
Test not performed	9	-	9 (13.0)	

**Table 2 cancers-17-01314-t002:** Chemotherapy regimens used in each line of treatment.

		n	% (n/N × 100%)	95% CI Inf	95% CI Sup
Chemotherapy regimen: relapse 1 N = 100	Carboplatin + paclitaxel	74	74.00%	64.27%	82.26%
	Carboplatin + gemcitabine	9	9.00%	4.20%	16.40%
	Topotecan	8	8.00%	3.52%	15.16%
	Doxorubicin pegylated liposomal	7	7.00%	2.86%	13.89%
	Paclitaxel	1	1.00%	0.03%	5.45%
	Gemcitabine	1	1.00%	0.03%	5.45%
Chemotherapy regimen: relapse 2 N = 48	Carboplatin + paclitaxel	26	54.17%	39.17%	68.63%
	Gemcitabine	8	16.67%	7.48%	30.22%
	Topotecan	7	14.58%	6.07%	27.76%
	Liposomal pegylated doxorubicin	4	8.33%	2.32%	19.98%
	Etoposid	2	4.17%	0.51%	14.25%
	Paclitaxel	1	2.08%	0.05%	11.07%
Chemotherapy regimen: relapse 3 N = 20	Gemcitabine	8	40.00%	19.12%	63.95%
	Liposomal pegylated doxorubicin	4	20.00%	5.73%	43.66%
	Topotecan	3	15.00%	3.21%	37.89%
	Carboplatin + gemcitabine	2	10.00%	1.23%	31.70%
	Etoposid	1	5.00%	0.13%	24.87%

## Data Availability

The dataset is available on request from the authors.

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
