# Peer review of "Enhancing Prognosis in Advanced Ovarian Cancer: Primary Cytoreductive Surgery and Adjuvant Chemotherapy or Neoadjuvant Chemotherapy and Interval Cytoreduction—A Single-Center Retrospective Observational Study"

_cancers, 2025, doi:10.3390/cancers17081314_

Round 1
Reviewer 1 Report
Comments and Suggestions for Authors
Gheorghe AS et al presented a retrospective research article aimed at evaluating the efficacy of two different treatment approaches for the management of advanced ovarian cancer patients. Specifically, they compared the outcome of primary cytoreductive surgery and adjuvant chemotherapy vs the primary cytoreductive surgery and neoadjuvant chemotherapy. They also take into account the contribution of BRCA mutations in the outcomes of patients. Overall, the topic covered by the authors is very interesting, however, there are some minor/major issues that they have to address before publication:
1) In the methods section of the Abstract, please clearly indicate the number of patients included in your study;
2) Please provide supporting references for the following statement: “Therefore, according to ESMO-ESGO guidelines, it is important that the selection of patients who are appropriate for primary debulking surgery is carried out in a specialised ovarian cancer center as the decision-making process typically involves a multidisciplinary evaluation that weighs the risks and benefits of the procedure based on individual patient character- 107 istics.”. For this purpose, please see:
- https://doi.org/10.3892/ijo.2021.5233
- https://doi.org/10.1136/ijgc-2023-004563
- https://doi.org/10.1007/s00404-023-07368-z
3) Since you presented your manuscript as an original article, please avoid to divide the Introduction section into different subheadings. In addition, you have to reduce the length of each section. In the present form, the introduction section is more appropriate for a review article and not an original one;
4) Check grammar of the following sentence: “The idea of neoadjuvant chemotherapy (NACT) developed as a response to several limitations associated with the standard treatment for advanced stages.”;
5) Chapter 1.5 should be removed, while chapter 1.6 needs to be shortened;
6) Change the word “Imagistically” in the methods section;
7) Provide a graphical representation of the different treatments investigated or the inclusion and exclusion criteria adopted (e.g. an algorithm or a workflow);
8) In tables 1 and 2, please perform statistics assessing the presence of differences between groups considering the anthropometric and pathological parameters;
9) Table 4 contains several data already reported in table 1 and 2. Please avoid redundancy;
10) Are there any statistical differences related to the data reported in Table 6?; Please, clarify;
11) Please report the non-significant Kaplan-Meier Curve as supplementary Figures.
Author Response
We sincerely thank you for the time and effort invested in evaluating our manuscript. We highly appreciate your constructive comments and insightful suggestions, which have significantly improved the quality and clarity of our work. Below, we provide detailed responses to each point raised. For clarity, our answers are inserted directly under each comment.
Gheorghe AS et al presented a retrospective research article aimed at evaluating the efficacy of two different treatment approaches for the management of advanced ovarian cancer patients. Specifically, they compared the outcome of primary cytoreductive surgery and adjuvant chemotherapy vs the primary cytoreductive surgery and neoadjuvant chemotherapy. They also take into account the contribution of BRCA mutations in the outcomes of patients. Overall, the topic covered by the authors is very interesting, however, there are some minor/major issues that they have to address before publication:
1) In the methods section of the Abstract, please clearly indicate the number of patients included in your study;
We thank you for pointing this out. We have now explicitly added the total number of patients included in the study (n = 100) in the Methods section of the Abstract.
2) Please provide supporting references for the following statement: “Therefore, according to ESMO-ESGO guidelines, it is important that the selection of patients who are appropriate for primary debulking surgery is carried out in a specialised ovarian cancer center as the decision-making process typically involves a multidisciplinary evaluation that weighs the risks and benefits of the procedure based on individual patient character- 107 istics.”. For this purpose, please see:
- https://doi.org/10.3892/ijo.2021.5233
- https://doi.org/10.1136/ijgc-2023-004563
- https://doi.org/10.1007/s00404-023-07368-z
We appreciate the reviewer’s suggestion and have now included the recommended references to support this important statement. Specifically, the citation has been updated and referenced as [15] in the revised manuscript.
3) Since you presented your manuscript as an original article, please avoid to divide the Introduction section into different subheadings. In addition, you have to reduce the length of each section. In the present form, the introduction section is more appropriate for a review article and not an original one;
We thank the reviewer for this valuable observation. In response, we have thoroughly revised the Introduction section to better align with the format and expectations of an original article. Specifically, we have removed all subheadings to improve the narrative flow and significantly reduced the length of the introduction by focusing on key clinical concepts directly relevant to our study objectives.
To further streamline the section and avoid a review-style structure, we have deleted the following paragraphs entirely:
“Molecular Biomarkers and Assessing the Efficacy of NACT”
“Hyperthermic Intraperitoneal Chemotherapy in Ovarian Cancer”
“Maintenance Therapy: Evolution of PARP Inhibitors – Past, Present and Future”
These changes help emphasize the core clinical and methodological aspects of our original retrospective research, and we trust the revised version now fits the intended article type more appropriately.
4) Check grammar of the following sentence: “The idea of neoadjuvant chemotherapy (NACT) developed as a response to several limitations associated with the standard treatment for advanced stages.”;
Thank you for pointing this out. We have rephrased the sentence for improved clarity and grammatical correctness. The revised sentence now reads:
“Neoadjuvant chemotherapy (NACT) was proposed as an alternative to address the shortcomings of primary surgery in patients with advanced ovarian cancer.
5) Chapter 1.5 should be removed, while chapter 1.6 needs to be shortened;
We have removed both sections entirely from the manuscript, as their detailed content was not essential to the core focus of the study.
6) Change the word “Imagistically” in the methods section;
We have revised this phrasing for clarity and academic appropriateness. The term has been replaced with “radiologically suspected,” which is more accurate and commonly accepted in scientific literature.
7) Provide a graphical representation of the different treatments investigated or the inclusion and exclusion criteria adopted (e.g. an algorithm or a workflow);
As requested, we have included a flowchart summarizing the study design, inclusion/exclusion criteria, and treatment allocation strategies to enhance the reader’s understanding of the research framework.
8) In tables 1 and 2, please perform statistics assessing the presence of differences between groups considering the anthropometric and pathological parameters;
9) Table 4 contains several data already reported in table 1 and 2. Please avoid redundancy;
Thank you for these observations. To reduce redundancy and enhance clarity, we have merged the relevant content from Tables 1 and 2 into Table 4, which now includes all comparative statistics between the treatment groups, including anthropometric and pathological parameters.
10) Are there any statistical differences related to the data reported in Table 6?; Please, clarify;
We appreciate this comment. Upon review, we concluded that Table 6 did not present statistically significant differences between groups. As such, we have removed the table and incorporated a brief summary of survival data in the text, noting the lack of statistically significant results.
11) Please report the non-significant Kaplan-Meier Curve as supplementary Figures.
We have followed this recommendation and moved the Kaplan-Meier survival curves without statistically significant results (e.g., PFS1, PFS2, OS) to the Supplementary Material section, where they are appropriately referenced in the main text.
Reviewer 2 Report
Comments and Suggestions for Authors
The authors present a retrospective singe centre real life analysis of patients with primary advanced high grade ovarian cancer with an emphasis on survival outcomes between NACT and primary debulking surgery. In general, the manuscript is written in understandable way.
I would suggest to shorten the introduction as the majority of the text in the introduction should be placed in the discussion. Maybe an editor could comment on thid.
I would suggest that the authors confirm if I understand inclusion/exclusion criteria correctly. Inclusion criteria says: Preoperative imaging ruling out unresectable disease according to ESGO criteria. Does this mean that in their centre the patients that have resectable disease, ECOG status 0-1, are offered NACT? Furthermore, I would be interested in where the disease was spread among patients having had FIGO stage IVB? As they were often treated with primary debulking surgery? The authors do not present the data about how many times the surgery was ended as complete without rezidual disease. The authors did not include the data about the influence of complete resection of the disease to the survival.
The description of covid-19 influence on treatment modality and delays should be placed under the discussion and not under the result section.
The results could be written in shorter way, without duplicating the data in the text and in the tables/graphs. In the table 4 I would suggest to describe the localisation of the disease among patients being in FIGO IV stage.
Author Response
We sincerely thank you for the time and effort invested in evaluating our manuscript. We highly appreciate your constructive comments and insightful suggestions, which have significantly improved the quality and clarity of our work. Below, we provide detailed responses to each point raised. For clarity, our answers are inserted directly under each comment.
The authors present a retrospective singe centre real life analysis of patients with primary advanced high grade ovarian cancer with an emphasis on survival outcomes between NACT and primary debulking surgery. In general, the manuscript is written in understandable way.
I would suggest to shorten the introduction as the majority of the text in the introduction should be placed in the discussion. Maybe an editor could comment on thid.
We have substantially revised the Introduction section by reducing its length and removing subheadings, in order to avoid overlap with the Discussion and to better fit the format of an original article.
I would suggest that the authors confirm if I understand inclusion/exclusion criteria correctly. Inclusion criteria says: Preoperative imaging ruling out unresectable disease according to ESGO criteria. Does this mean that in their centre the patients that have resectable disease, ECOG status 0-1, are offered NACT?
Thank you for this important question. In our centre, the treatment decision is made by a multidisciplinary team, and NACT is not offered by default to all patients with resectable disease and ECOG 0–1. However, we would like to clarify that some of the patients included in our cohort were initially referred from non-specialized gynecologic or general surgical units, where the surgical expertise in advanced ovarian cancer debulking is more limited. In such cases, the referring teams may opt to initiate neoadjuvant chemotherapy before transferring the patient to a tertiary centre for interval cytoreductive surgery. Additionally, in certain instances, delays in surgical scheduling, due to high operating room demand or resource constraints, can lead to a decision to initiate systemic treatment (NACT) to avoid disease progression during the waiting period. These real-life situations reflect the complexity of treatment pathways in the current healthcare system and underscore the value of retrospective studies in capturing such variability.
Furthermore, I would be interested in where the disease was spread among patients having had FIGO stage IVB? As they were often treated with primary debulking surgery?
We have now added a paragraph in the Results section providing details on the disease localization among patients with FIGO stage IVB.
The authors do not present the data about how many times the surgery was ended as complete without rezidual disease. The authors did not include the data about the influence of complete resection of the disease to the survival.
We acknowledge the importance of reporting complete cytoreduction (R0) rates and their correlation with survival. However, we regret to inform that this specific information was not consistently documented in the surgical or histopathological reports from 2018 for a significant proportion of patients. Due to the retrospective nature of the study and the incomplete data availability, we were unable to reliably include this parameter in our analysis. We have now clarified this limitation in the revised manuscript and also removed it from inclusion criteria (we aimed to assess it at first).
The description of covid-19 influence on treatment modality and delays should be placed under the discussion and not under the result section.
We agree with the reviewer and have moved the analysis related to the influence of the COVID-19 pandemic from the Results section to the Discussion.
The results could be written in shorter way, without duplicating the data in the text and in the tables/graphs. In the table 4 I would suggest to describe the localisation of the disease among patients being in FIGO IV stage.
We reviewed the entire Results section and removed redundant text that duplicated information already presented in the tables and figures. Table 4 is the only one left to avoid overlap with Tables 1 and 2. In addition, localization data for FIGO IVB patients has been explicitly described, as recommended.
Reviewer 3 Report
Comments and Suggestions for Authors
- The study is retrospective, but the rationale for this design is not explicitly stated. Given the availability of randomized controlled trial (RCT) data on PCS vs. NACT-IDS, the authors should discuss why a retrospective study adds new insights.
- Were there any institutional biases or selection criteria that may have influenced treatment allocation? A discussion of potential confounders is necessary.
- The inclusion and exclusion criteria are well defined, but the manuscript does not provide a comparative table of baseline characteristics between the two treatment groups.
- Differences in patient selection, tumor burden, or clinical performance could significantly impact survival outcomes and should be explicitly addressed.
- How were BRCA testing and homologous recombination deficiency (HRD) assessments performed? Were they uniformly applied to both groups?
- The manuscript reports that 47% of patients received Bevacizumab as maintenance therapy, but Olaparib was not available at the time.
- Given the significant role of PARP inhibitors in BRCA-mutated and HRD-positive patients, the study should discuss how this limitation may have affected outcomes.
- The subgroup analysis comparing Bevacizumab vs. Olaparib at first relapse suggests improved OS with Olaparib, but the sample size is small (46 vs. 24 patients). This should be interpreted cautiously.
- The manuscript does not clearly state the next steps based on these findings. Would a prospective trial be feasible?
- Could molecular biomarkers (e.g., KELIM score, HRD status) be integrated into future treatment algorithms?
Author Response
We sincerely thank you for the time and effort invested in evaluating our manuscript. We highly appreciate your constructive comments and insightful suggestions, which have significantly improved the quality and clarity of our work. Below, we provide detailed responses to each point raised. For clarity, our answers are inserted directly under each comment.
The study is retrospective, but the rationale for this design is not explicitly stated. Given the availability of randomized controlled trial (RCT) data on PCS vs. NACT-IDS, the authors should discuss why a retrospective study adds new insights.
We appreciate this important observation. Although multiple RCTs comparing primary cytoreductive surgery (PCS) and neoadjuvant chemotherapy with interval debulking surgery (NACT-IDS) are already available, our retrospective study offers added value by reflecting the real-world treatment patterns and limitations encountered in a Romanian oncologic care setting. These include variability in access to specialized surgical teams, institutional logistics that may delay timely surgery, limited availability of molecular testing or targeted therapies (e.g., PARP inhibitors), and differences in maintenance strategies. In such settings, a retrospective design allows for the analysis of treatment outcomes across a more heterogeneous population, which may not be represented in strictly controlled trial settings.
We have now included a statement in the Discussion section clarifying the rationale behind the retrospective design and the relevance of our findings for healthcare systems with similar constraints.
Were there any institutional biases or selection criteria that may have influenced treatment allocation? A discussion of potential confounders is necessary.
Yes, we acknowledge that treatment allocation was not randomized and may have been influenced by several institutional and logistical factors. For example, some patients were referred from non-specialized centers that opted for NACT due to limited surgical expertise, while others were initiated on chemotherapy due to long waiting times for surgery. These practical constraints, along with clinical considerations such as comorbidities or tumor burden, could have introduced selection bias.
Moreover, due to the absence of a national cancer registry or institutional ovarian cancer database, patient selection for this retrospective study was performed manually. We reviewed and included consecutive patient files diagnosed with advanced high-grade serous ovarian cancer, available in the hospital’s medical archive, until reaching a predefined sample size of 100 cases. This method, while pragmatic, may also contribute to the heterogeneity of the cohort and represents an inherent limitation of retrospective research in our context.
The inclusion and exclusion criteria are well defined, but the manuscript does not provide a comparative table of baseline characteristics between the two treatment groups. Differences in patient selection, tumor burden, or clinical performance could significantly impact survival outcomes and should be explicitly addressed.
Thank you for pointing this out. There is a comprehensive comparative table (revised Table 1) presenting baseline characteristics of the two treatment groups, including statistical comparisons between groups, and the potential influence of these differences on survival outcomes has been discussed in the Results and Discussion sections.
How were BRCA testing and homologous recombination deficiency (HRD) assessments performed? Were they uniformly applied to both groups?
At the time the study cohort was treated initially, neither BRCA nor HRD testing was routinely performed for all patients, due to limited access and lack of reimbursement. BRCA testing became more widely available only after it started to be supported by the Astra Zeneca, following its approval and reimbursement in Romania. Similarly, HRD testing became accessible more recently, after the reimbursement of the bevacizumab-olaparib combination based on the PAOLA-1 trial. As a result, comprehensive genomic profiling was not uniformly applied across the cohort. However, some long-term survivors underwent molecular testing retrospectively, either as part of their follow-up care or by self-funding these analyses at the time of diagnosis.
The manuscript reports that 47% of patients received Bevacizumab as maintenance therapy, but Olaparib was not available at the time.
Given the significant role of PARP inhibitors in BRCA-mutated and HRD-positive patients, the study should discuss how this limitation may have affected outcomes.
Indeed, the unavailability of Olaparib as maintenance therapy during the period of this study is a major limitation. This likely impacted the long-term outcomes, particularly in patients with BRCA mutations or HRD-positive status, who would have otherwise benefited from PARP inhibitor therapy. We have now added a statement in the Discussion section acknowledging this limitation and suggesting that the observed survival results may underestimate the potential benefits in a modern therapeutic context.
The subgroup analysis comparing Bevacizumab vs. Olaparib at first relapse suggests improved OS with Olaparib, but the sample size is small (46 vs. 24 patients). This should be interpreted cautiously.
We fully agree with the reviewer. The subgroup analysis comparing outcomes at first relapse was exploratory and based on a small sample size. These results should be interpreted with caution and are not intended to draw definitive conclusions. We have now explicitly mentioned this limitation in the Discussion, emphasizing the need for further studies to validate these findings.
The manuscript does not clearly state the next steps based on these findings. Would a prospective trial be feasible?
Could molecular biomarkers (e.g., KELIM score, HRD status) be integrated into future treatment algorithms?
Thank you for this excellent suggestion. Based on our findings and the challenges identified in real-life practice, we really aim to participate in a prospective study focusing on individualized treatment selection, guided by molecular biomarkers such as HRD status and dynamic tools like the KELIM score, could significantly optimize outcomes for patients with advanced ovarian cancer. We have added a paragraph in the Conclusion outlining this perspective and encouraging the integration of biomarker-driven strategies in future research protocols.
Round 2
Reviewer 1 Report
Comments and Suggestions for Authors
The authors well-addressed all of my previous comments. The manuscript is now more detailed and clear and can be accepted for publication.
Reviewer 2 Report
Comments and Suggestions for Authors
Thank you very much for all the effort and clarifications.